# Comprehensive Review: Unveiling the Pro-Oncogenic Roles of IL-1ß and PD-1/PD-L1 in NSCLC Development and Targeting Their Pathways for Clinical Management

**DOI:** 10.3390/ijms241411547

**Published:** 2023-07-17

**Authors:** Dani Ran Castillo, Won Jin Jeon, Daniel Park, Bryan Pham, Chieh Yang, Bowon Joung, Jin Hyun Moon, Jae Lee, Esther G. Chong, Kiwon Park, Mark E. Reeves, Penelope Duerksen-Hughes, Hamid R. Mirshahidi, Saied Mirshahidi

**Affiliations:** 1Division of Hematology and Oncology, Loma Linda University Cancer Center, Loma Linda, CA 92354, USA; dran@llu.edu (D.R.C.); egchong@llu.edu (E.G.C.); mereeves@llu.edu (M.E.R.); hmirshah@llu.edu (H.R.M.); 2Department of Internal Medicine, Loma Linda University, Loma Linda, CA 92350, USA; wjjeon@llu.edu (W.J.J.); bpham@llu.edu (B.P.); bjoung@llu.edu (B.J.); jmoon1@llu.edu (J.H.M.); 3Department of Internal Medicine, University of San Francisco-Fresno, Fresno, CA 93701, USA; daniel.park2@tu.edu; 4Department of Internal Medicine, School of Medicine, University of California Riverside, Riverside, CA 92521, USA; chiehy@medsch.ucr.edu; 5School of Medicine, Loma Linda University, Loma Linda, CA 92350, USA; jlee36@students.llu.edu; 6Department of Pharmacy, Loma Linda University, Loma Linda, CA 92350, USA; kipark@llu.edu; 7Division of Biochemistry, Department of Medicine & Basic Sciences, School of Medicine, Loma Linda University, Loma Linda, CA 92350, USA; pdhughes@llu.edu; 8Biospecimen Laboratory, Loma Linda University Cancer Center, Loma Linda, CA 92354, USA; 9Division of Microbiology and Molecular Genetics, Department of Medicine & Basic Sciences, Loma Linda University, Loma Linda 92350, CA, USA

**Keywords:** interleukin-1 beta (IL-1β), immune-checkpoint inhibitors (ICIs), non-small cell lung cancer (NSCLC), programmed death ligand 1 (PD-L1), therapeutic resistance

## Abstract

In the past decade, targeted therapies for solid tumors, including non-small cell lung cancer (NSCLC), have advanced significantly, offering tailored treatment options for patients. However, individuals without targetable mutations pose a clinical challenge, as they may not respond to standard treatments like immune-checkpoint inhibitors (ICIs) and novel targeted therapies. While the mechanism of action of ICIs seems promising, the lack of a robust response limits their widespread use. Although the expression levels of programmed death ligand 1 (PD-L1) on tumor cells are used to predict ICI response, identifying new biomarkers, particularly those associated with the tumor microenvironment (TME), is crucial to address this unmet need. Recently, inflammatory cytokines such as interleukin-1 beta (IL-1β) have emerged as a key area of focus and hold significant potential implications for future clinical practice. Combinatorial approaches of IL-1β inhibitors and ICIs may provide a potential therapeutic modality for NSCLC patients without targetable mutations. Recent advancements in our understanding of the intricate relationship between inflammation and oncogenesis, particularly involving the IL-1β/PD-1/PD-L1 pathway, have shed light on their application in lung cancer development and clinical outcomes of patients. Targeting these pathways in cancers like NSCLC holds immense potential to revolutionize cancer treatment, particularly for patients lacking targetable genetic mutations. However, despite these promising prospects, there remain certain aspects of this pathway that require further investigation, particularly regarding treatment resistance. Therefore, the objective of this review is to delve into the role of IL-1β in NSCLC, its participation in inflammatory pathways, and its intricate crosstalk with the PD-1/PD-L1 pathway. Additionally, we aim to explore the potential of IL-1β as a therapeutic target for NSCLC treatment.

## 1. Introduction

Lung cancer remains the leading cause of cancer-related mortality and the second most common malignancy worldwide [1]. Currently, in NSCLC, initial management begins with searching for targetable drivers such as *EGFR*, *ALK*, *ROS-1*, *NTRK*, *RET*, *NRG* or *BRAF, MET*; however, for patients without such mutations, treatment options are limited [2]. Recent studies have demonstrated that the TME of NSCLC involves a proinflammatory state that can be targeted [3]. Dysregulated inflammatory conditions contribute to lung carcinogenesis by causing protumor inflammation, tumor suppression, and the activation of oncogenes [1,4]. Recently, the Canakinumab Anti-inflammatory Thrombosis Outcome Study (CANTOS) trial incidentally demonstrated that inhibiting IL-1β is associated with a significantly reduced incidence and mortality of lung cancer [5]. Therefore, proposing IL-1β, a key inflammatory cytokine, is a targetable area that directly affects the proinflammatory nature of NSCLC development and progression [5]. TME is becoming recognized as a key factor in carcinogenesis and is a topic of great interest for seeking management options for NSCLC without targetable mutations. In the TME, overexpression of the PD-L1 on tumor cells and engagement with its receptor programmed cell death protein 1 (PD-1) causes T-cell exhaustion and results in decreased anti-tumor activity [6,7]. Immunomodulators targeting the PD-1/PD-L1 axis have had promising results, but therapeutic resistance is common and challenging for clinicians. Here, the role of IL-1β inhibitors is an area of great interest as preclinical and clinical studies have demonstrated that increased levels of IL-1β are associated with elevated expression of PD-L1 and decreased response to anti-PD-L1 therapy [8,9].

Recent advances in understanding the IL-1β/PD-1/PD-L1 pathway have prompted attempts to utilize combinatorial strategies that involve IL-1β inhibitors together with ICIs. To identify biomarkers that predict the ICI response in tumors without actionable mutations and provide a solution for patients with actionable mutations who develop resistance to currently approved targeted therapies, understanding the crosstalk that occurs within the IL-1β/PD-1/PD-L1 pathway and the TME alterations that arise when ICIs are combined with IL-1β inhibitors is crucial. Therefore, in this review, we aim to discover the role of IL-1β in NSCLC, its involvement in inflammatory pathways, and its crosstalk with the PD-1/PD-L1 pathway as well as explore its potential role in the treatment of NSCLC. We hope to shed light on the use of combination therapies for NSCLC without targetable mutations and provide a potential treatment option for NSCLC patients.

## 2. Il-1β Signaling in NSCLC

The TME is altered by many immune factors, and recent studies have shown that the TME of lung cancer is proinflammatory [10]. Due to the relationship between chronic inflammatory states and cancer, serum levels of IL-1β, IL-6, IL-8 [11] have been studied as potential biomarkers of malignancy. For example, patients with breast, lung, cervical, hepatocellular, and gastric cancer tumor cells revealed increased levels of IL-1β alleles [12]. In general, the expression of IL-1β is induced by the presence of stressful stimuli such as hypoxia, inflammation, infection, which induces toll-like receptors (TLRs) to stimulate tumor necrosis factor (TNF). Inactive pro-IL1β is activated by intracellular protein complexes, known as inflammasomes, and then cleaved by caspase-1 (also known as IL-1β-converting enzyme (ICE)) into its active form. Immune cells involved in tumor suppression, T cells, dendritic cells (DCs), epithelial cells, neutrophils, macrophages, and other antigen-presenting cells (APCs) express IL-1 receptors (IL-1R and IL-1R2) which interact with IL-1β (Figure 1). IL-1β signaling recruits myeloid differentiation primary response-88 (MyD88) and IL-1R associated kinases (IRAKs), which directly interact with TNF receptor-associated factor 6 (TRAF6). This activates the mitogen-activated protein kinase (MAPK) pathway and (nuclear factor kappa B) NF-κB, thus activating downstream inflammatory pathways and promoting the unchecked growth of malignant cells and inhibiting apoptosis [8,12,13].

Chronic inflammation has been linked to an increased risk of cancer by providing survival signals, suppressing T-cells’ effector functions, inducing angiogenesis, and promoting invasion and metastasis [3,14]. The active form of IL-1β induces protumor inflammation, tumorigenesis, immune invasion, and metastasis (Figure 2) [3,4,13]. IL-1β may also have a positive feedback loop, perpetuating the course of the cancer [12]. Within the TME, tumor cells are the source of chronic inflammation, and IL-1β becomes an upstream regulator, altering immunologic responses, hormonal signaling, neovascularization, and enhancing metastatic potential through downstream pathways such as NF-κB/MAPK/Protein Kinase B (AKT)/Wnt/β-catenin [13].

The role of IL-1β in cancer is complex, and its ability to directly alter the TME has been studied in the preclinical setting. For example, an increased expression of IL-1β has been shown to promote the accumulation of myeloid-derived suppressor cells (MDSCs), decrease the number of natural killer (NK) cells, and increase tumor size [4,15]. The levels of MDSCs, our body’s major suppressors of immunological responses to tumors, are significantly increased in the TME, which leads to the maintenance and acceleration of tumor growth and metastasis [4]. An increased accumulation of MDSCs results in the expansion of CD4^+^ CD25^+^ Foxp3^+^ regulatory T cells (Tregs), leading to downregulation of the antitumor capability of NK and cytotoxic T cells. IL-1β also functions as a chemoattractant to recruit tumor-associated macrophages (TAMs) by attaching monocyte chemoattractant protein (MCP-1) on tumor cells [16]. This cascade further potentiates IL-1β production via TAMs and triggers inflammasomes to promote further tumor growth [15].

Furthermore, IL-1β has been implicated in the promotion of angiogenesis in several types of malignancies, such as melanoma and fibrosarcoma, potentiating their metastatic potential [12,17,18]. This is due to its ability to induce the expression of pro-angiogenic factors such as vascular endothelial growth factor (VEGF), basic fibroblast growth factor (bFGF), transforming growth factor (TGF-αβ), platelet-derived endothelial growth factor (PDGF), interleukin-8 (IL-8, also known as CXCL8), and fibroblast growth factors (FGF). They mediate the migration and proliferation of endothelial cells to expand the existing vascular system, which is necessary for the delivery of nutrients and oxygen to rapidly growing tumor cells [17]. IL-1β also increases the expression of matrix metalloproteinases (MMPs), which degrade extracellular matrix (ECM) proteins, facilitating the migration and invasion of tumor cells into surrounding tissues [3,17]. MMPs further promote angiogenesis by releasing sequestered pro-angiogenic factors from the ECM, making them available to surrounding cells. In addition, in lung cancer, exposure to IL-1β decreases phosphatase and tensin homolog (PTEN) expression, phosphoinositide 3-kinase (PI3K)/AKT signaling activation, and the induction of epithelial–mesenchymal transition (EMT), conferring primary lung carcinoma cells with the ability to mobilize, invade, and damage distant sites, leading to angiogenesis [9,19,20].

Specifically, for NSCLC, IL-1β has a multifaceted impact on the development and progression of the disease. Patients with NSCLC have been found to have elevated levels of IL-1β in plasma and IL-1β mRNA expression, and serum IL-1β has been associated with worse prognosis [14,21,22]. In NSCLC, IL-1β potentiates accelerated neoplastic progression by repressing miR-101 expression through the cyclooxygenase 2 (COX2)/HIF1α pathway [21]. Tumor-derived IL-1β activates γδT-cells, which in turn secrete IL-17. The secretion of IL-17 causes neutrophils to increase inducible nitric oxide synthase (iNOS) and suppress anti-tumor CD8^+^ T cells, leading to an increase in metastatic potential [23]. In lung cancer, it has also been shown that commensal bacteria promote IL-1β production from local macrophages, inducing tumor cell proliferation and inflammation [24].

Despite recent advances in targeted therapies for NSCLC patients with specific genetic mutations, the lack of robust response has been a challenge for clinicians [25]. IL-1β has been implicated in the development of therapeutic resistance. For example, it has been observed that varying levels of IL-1β are present in therapy-resistant patients [3,26]. Thus, the heterogeneity of serum IL-1β levels may explain the variability of response to treatment options in NSCLC [26,27]. Additionally, mutations found in lung cancer cells are associated with the IL-1β axis, making it a potential target for therapy [8]. *BRAF (V600E)* mutations are rare in NSCLC but can occur in up to 2% of cases [28]. Treatment of these neoplasms with BRAF inhibitors (BRAFi) has been shown to increase dendritic cell-mediated IL-1β production, worsening the inflammatory positive feedback loop and promoting resistance. Hajek et al. demonstrated that the combination of dabrafenib with vemurafenib or trametinib, BRAFis used in melanoma and NSCLC patients with BRAFV600E mutation, strongly upregulated IL-1β production in myeloid mouse APCs due to BRAFi-induced activation of the inflammasome leading to caspase-8 activation and pro-IL-1β processing. Alternative mechanisms explain BRAFi resistance by a cytokine-signaling network involving TAM-derived IL-1β, cancer-associated fibroblasts (CAFs)-derived CXCR2 ligands, and PTEN inactivation [29]. Still, the mechanism of resistance to BRAFi and BRAFi combinatorial regimens remains an unmet area of further study. Given that IL-1β production affects PTEN, it is reasonable to hypothesize that IL-1β inhibitors may aid in combating resistance to BRAFi. Therefore, further studies are warranted to explore the potential of IL-1β inhibitors in combination with BRAFi in NSCLC patients with BRAF mutations.

The role of IL-1β in the development of resistance to treatment in NSCLC is further implicated in current studies with bortezomib and EGFR inhibitors. Davies et al. showed that bortezomib, a proteasome inhibitor that partially inhibits NF-κB, is not effective as a single agent in the treatment of NSCLC [30]. Further investigation revealed that the inhibition of NF-κB signaling was associated with an increase in IL-1β production and enhanced tumorigenesis in the lungs. Interestingly, McLeod et al. demonstrated that dual blockade with an IL-1R antagonist (anakinra) and bortezomib resulted in increased therapeutic efficacy by reducing lung tumor burden [31]. These findings suggest that the inefficiency of monotherapy with NF-κB inhibitors in NSCLC may be due to the potentiation of neutrophil-dependent production of IL-1β, leading to enhanced pulmonary carcinogenesis. The study also highlights the role of IL-1β in resistance to currently available treatment modalities and suggests that combined NF-κB and IL-1β targeted treatments may lead to reduced tumor formation and growth [16,31]. The recent study conducted by Yuan et al. revealed that blocking the IL-1β pathways resulted in an upsurge in cytotoxic CD8+ T cell infiltration and a reduction in the protumor immunosuppressive response. Additionally, this treatment proved to be highly effective in inhibiting the activation of the NF-κB and STAT3 pathways [32].

In addition, Huang et al. found that IL-1β upregulates EH domain-containing protein 1 (EHD1) expression, which activates the PTEN/PI3K/AKT signaling pathway, leading to off-site EGFR-TKI resistance in NSCLC. Interestingly, the inhibition of the IL-1β/EHD1/TUBB3 axis has shown promising results in overcoming EGFR-TKI resistance [9]. These findings suggest that inhibition of the IL-1β/EHD1 signaling pathway may be a target for patients who develop EGFR-TKI resistance. Overall, the combinatorial, targeted strategies toward the IL-1β axis may be a solution for NSCLC patients with actionable mutations who have developed resistance to currently approved targeted therapies.

## 3. Interplay of PD-1/PD-L1 with TME and Their Role in Treatment Resistance

Among targeted therapies for NSCLC, tyrosine kinase inhibitors (TKIs) against specific driver-mutations have shown significant improvement in survival outcomes [33]. However, only a subset of patients with NSCLC carries single-driver mutations against EGFR, ALK, ROS-1, or BRAF, while the majority lack such targetable mutations [33,34]. This has been a major challenge for the management of NSCLC. While novel treatment options for NSCLC with driver-mutations are standard of care, some patients develop resistance to such treatments, requiring another line of therapy [34]. In patients with NSCLC without targetable driver mutations and those with resistance to TKI therapy, the use of ICIs targeting the PD-1/PD-L1 pathway has been a topic of focused studies [35].

Preclinical studies have shown that the PD-1/PD-L1 pathway is interconnected with the TME [36,37]. PD-1 is expressed on T cells, specifically on tumor-specific T, B, and NK cells [37]. PD-L1 is usually expressed by macrophages, DCs, and epithelial cells along with some activated T and B cells and upregulated in certain tumor cells [38]. The interaction of PD-1 on T-cells and PD-L1 on tumor cells causes the phosphorylation of Lck, a tyrosine kinase, leading to T-cell deactivation and subsequently the compromised anti-tumor activity of T cells [39]. PD-L1 expression is a critical component of the adaptive immune mechanism of tumor cells to escape anti-tumor cells [40,41]. The interaction of PD-1/PD-L1 proteins in the TME results in T cell exhaustion, suppressing the anti-tumor effects of cells such as APCs and tumor infiltrating lymphocytes (TILs), leading to a state of dysfunction characterized by decreased T cell effector activity [37]. An “active” TME consists of the presence of PD-1^+^ CD8^+^ activated T cells and the expression of PD-L1 [36], which is a favorable setting for ICIs.

ICIs like pembrolizumab (PD-1 inhibitor), atezolizumab (PD-L1 inhibitor), cemiplimab (PD-L1 inhibitor), and durvalumab (PD-L1 inhibitor) have shown promising results in improving overall survival (OS) and progression-free survival (PFS) in certain types of NSCLC patients with tolerable adverse effects (Appendix A). Prominent clinical trials involving ICIs include KEYNOTE-010 and KEYNOTE-024, which showed improved survival with pembrolizumab, and CheckMate 017 and CheckMate 057, which reported longer survival with nivolumab [26,42,43,44,45]. Numerous studies have explored the use of ICIs for the treatment of various settings in NSCLC. In EMPOWER-Lung 1, cemiplimab was used as a first-line monotherapy for advanced NSCLC with at least 50% PD-L1 expression, and it showed significant improvement in OS and PFS compared to platinum-doublet chemotherapy [46]. Similarly, durvalumab after chemoradiotherapy in stage III NSCLC in PACIFIC showed significantly longer PFS compared to placebo [47]. However, MYSTIC demonstrated that durvalumab as a first-line treatment for metastatic NSCLC did not improve OS [48]. In other studies, the combination of atezolizumab with chemotherapy in stage IV NSCLC showed improved PFS in Impower131 and significant improvements in OS and PFS in Impower130 for non-squamous NSCLC [49,50]. Similarly, atezolizumab also showed superior OS as a first-line treatment for metastatic NSCLC with at least 1% PD-L1 expression in Impower110, and it was added on to bevacizumab-carboplatin-paclitaxel for metastatic non-squamous NSCLC in Impower150 [51]. Additionally, the OAK trial revealed improved survival with atezolizumab in patients with previously treated NSCLC with low or undetectable PD-L1 expression [52].

Despite these promising results, resistance to ICIs is a significant challenge [4,53,54]. The use of ICIs, specifically PD-1/PD-L1 inhibitors, has shown variable clinical benefit with major pathological regression rates ranging widely from 18 to 83% [8]. Recent studies have shown that resistance to anti-PD therapy may be one of explanations for this widely varied response [4,55,56]. Primary and acquired resistance occur from immune-mediated mechanisms which derive from the TME and can occur at any step of the immunoregulatory process [4,55,57,58]. Primary resistance is the lack of initial response to ICIs, whereas acquired resistance is the lack of further response after initial improvement with ICIs [59]. Some of the mechanisms of resistance to anti-PD therapy include immunologic loss or lack of neoantigens, signaling defects in the interferon (IFN) pathway, the lack of PD-L1 receptors, and local immune dysfunction, including the exclusion of T cells [55,60]. A key mechanism of resistance to ICIs, specifically anti-PD therapy, is the lack of PD-L1 receptors. Recently, the TME of malignancies such as NSCLC and melanoma has been classified into four categories based on the expression of PD-L1 and TILs [55,60]. Type 1 is TIL^−^ and PD-L1^−^, type 2 is TIL^+^ and PD-L1^+^, type 3 is TIL^+^ and PD-L1^−^ and type 4 is TIL^−^ and PD-L1^+^. Unfortunately, only about 17% of NSCLCs demonstrate type 2 TME, which may explain the unpredictable response to ICIs [55]. T cell infiltration is closely related to PTEN, and the loss of PTEN expression is associated with resistance to anti-PD therapy [61,62,63]. Further, though not fully understood, the gut microbiome may play a role in altering the TME by stimulating DCs and affecting antigen presentation [64,65]. Interestingly, sensitivity to anti-PD therapy has been associated with tumor mutational burden (TMB), with higher levels of mutations indicating high immunogenicity, which can be targets for anti-PD therapy [61,66]. However, studies have shown that some gene mutations can lead to mixed results to anti-PD therapy. For example, patients with positive EGFR mutation have been shown to have lower levels of PD-L1 expression with possible resistance to anti-PD therapy [67,68]. Such observations of resistance to ICIs may help guide their use in NSCLC, and further studies are needed.

In a retrospective study, Sakai et al. stratified NSCLC patients who had received ICIs therapy by morphological, immune, and genetic features to further address concerns about ICIs resistance in sub-groups [69]. The study revealed that the absence of definitive morphological features in non-squamous NSCLC was associated with better prognosis, whereas tumors diagnosed as morphologic adenocarcinoma were more associated with PD-L1 score < 1% and less likely to be associated with samples with PD-L1 expression > 50%. These results suggest the use of morphological features of NSCLC and levels of PD-L1 expression to predict response and resistance to ICIs [69]. IFN-γ has been shown to maintain levels of PD-L1, and the clinical significance of the IFN-γ pathway and other cytokines are areas requiring further evaluation [55,56]. Further, biomarkers such as microsatellite instability (MSI), high TMB along with CD8+ T cell infiltrates are noted as foremost predictive markers for ICIs response [70]. Understanding these mechanisms of resistance is crucial for improving the effectiveness of ICIs in NSCLC.

## 4. Complex Networking of IL-1ß and PD-1/PD-L1 Pathway and Implications for Targeted Therapies

The relationship between IL-1β and PD-1/PD-L1 is complex and may provide insight into overcoming the limitations and variable response to current therapies and the role of combination therapies of ICIs and interleukin inhibitors (Figure 2) [71]. Research involving hepatocellular carcinoma (HCC) [72,73,74], gastric cancer [75], melanoma [76], multiple myeloma [77] and other malignancies [78] demonstrated a positive association between IL-1β levels and PD-L1 expression. In a murine renal cell carcinoma (RCC) model, Aggen et al. showed that a combinatorial inhibition of IL-1β and PD-1 led to increased anti-tumor activity [79]. In a study of Kaposi’s sarcoma, Chen et al. found that silencing IL-1β by RNA inhibitors (RNAis) reduced PD-L1 expression [80]. Furthermore, targeting the IL-1β/PD-1/PD-L1 pathway can enhance the response to chemotherapy and radiation therapy, indicating its potential as a broad-spectrum cancer therapy [81].

Similarly, in NSCLC, studies have shown the fascinating interplay between IL-1β and PD-L1/ PD-1 [3,81]. According to a retrospective study by De Alencar et al., smoking history was associated with an increased production of IL-1β in combination with a higher expression of PD-L1 in NSCLC tumor cells [1]. The anti-tumor effects of co-inhibition of the IL-1β and PD-1/PD-L1 pathways may be due to dynamic changes in the TME. Li et al. demonstrated that acute IL-1β exposure led to chemoresistance and PD-L1 upregulation in the TME of NSCLC [81]. In a murine model of NSCLC, Kaplanov et al. demonstrated that combination therapy with both anti-IL-1β and anti-PD-1 treatment had a synergistic effect, leading to the complete inhibition of tumor growth for 30 days [82]. Tumor volume and weight were significantly reduced compared to the control group and either of the single-agent groups. The TME of tumor cells treated with the combination of anti-IL-1β and anti-PD-1 therapies showed a statistically higher percentage of CD8^+^ T cells, indicating a reversal of the immunosuppressive effects of IL-1β [82]. Similar findings were shown by Jayaraman et al., who demonstrated that canakinumab treatment in humanized NSCLC cells slowed tumor growth by remodeling the TME [83].

Studies have suggested a potential amplifying effect between PD-1 signaling and the immunosuppressive activity of MDSCs [84,85,86] and a positive correlation between IL-1β and the expansion of MDSCs [87]. The amplifying nature identified in the interaction between PD-1 and IL-1β with MDSCs, which serve as significant suppressors of immune activity, suggests a deeper interplay between these two signaling molecules in creating an immunosuppressive, carcinogenic TME. While ICIs as a standalone treatment for NSCLC have limited efficacy, studies suggest that IL-1β and PD-1/PD-L1 pathways may work synergistically as targets for NSCLC treatment due to the notable correlations and intersection between IL-1β and PD-1 expression (Figure 3) [88]. The inhibition of IL-1β may lead to the decreased release of dysregulated inflammatory cytokines, while simultaneous PD-1/PD-L1 inhibition targets T cells, overcoming their anti-tumor immunity [4]. IL-1β inhibitors have emerged as a promising treatment modality for NSCLC without targetable mutations. As discussed earlier, IL-1β promotes tumorigenesis by stimulating the production of inflammatory cytokines, promoting angiogenesis and invasion, and inhibiting apoptosis. Targeting IL-1β, therefore, has the potential to enhance the effectiveness of ICIs and other cancer treatments. The interconnectedness of the PD-1/PD-L1 and IL-1β pathways is not yet fully understood, but elucidating their relationship may provide promising avenues for NSCLC treatment [4].

## 5. IL-1β/PD-1/PD-L1 Pathways in Recent and Ongoing NSCLC Clinical Studies

Recent and ongoing trials have targeted the IL-1β/PD-1/PD-L1 pathway. Although CANOPY-1 did not meet its primary endpoints, further trials to evaluate this combinatorial approach are ongoing (Table 1) [89]. Similarly, CANOPY-2, a trial investigating the role of canakinumab in patients with NSCLC previously treated with PD-1/PD-L1 inhibitors and platinum-based chemotherapy, did not show significant improvement in PFS or OS in previously treated patients with advanced or metastatic NSCLC who received canakinumab with docetaxel compared to docetaxel alone [90]. Furthermore, there was a higher incidence of fatal infections, which was a concern seen in previous canakinumab trials. Although CANOPY-A, the latest trial of canakinumab, did not meet its primary endpoint of disease-free survival, its safety profile was comparable to previous trials [91]. Hence, while the relationship between IL-1β and PD-1/PD-L1 pathways in the TME of NSCLC is significant, further research is needed to elucidate the exact mechanisms and improve outcomes with IL-1β and PD-1/PD-L1 inhibitors.

There are ongoing clinical trials investigating the efficacy of anti-IL-1 strategies for NSCLC. These trials are examining the use of anti-IL-1 therapies either alone or in combination with chemotherapy or other immunotherapeutic agents (Table 2).

## 6. The Unmet Need for Predictive Biomarkers of Response and Resistance in NSCLC

Given the heterogeneity of response to ICIs and combination therapies with IL-1β and PD-1/PD-L1 inhibitors, there is an unmet need for predictive biomarkers of response and resistance. As stated above, PD-L1 expression is directly correlated with the function of IL-1β, and the elevated serum levels of IL-1β may portend improved response to ICIs. Elevated serum IL-1β levels were correlated with decreased response to PD-L1 treatment and worsened survival outcomes, alluding to the potential role for serum IL-1β as a biomarker of response to combination therapies [3]. In another study, Wang et al. noted the potential prognostic implication of tracking M-MDSCs-related genes (monocytic myeloid-derived suppressor cells) in the TME of lung adenocarcinoma (LUAD) [92]. Following the distinct genetic signatures of M-MDSCs-related genes may aid in our efforts to distinguish the effects of upregulated IL-1β and PD-1 expression in studying TME compositions that may have better responses to ICIs [92].

In the setting of the significance of the TME in the carcinogenesis and treatment of NSCLC along with the proinflammatory nature of NSCLC, studies have demonstrated the potential role of inflammatory cytokines and serum markers for predicting response to combination therapies. Levels of cytokines such as IL-6 and IL-8 may be relevant biomarkers for response to combined IL-1β and PD-1/PD-L1 inhibition [93,94]. Studies have shown a positive correlation between IL-6 and PD-1/PD-L1 expression, and in colorectal cancer, a strong correlation exists among IL-8, IL-β1, and MMP-2 [88,95]. In preclinical models of KRAS-mutant lung cancer, the use of IL-1β inhibitors has been shown to reduce tumor burden, increase the infiltration of CD8^+^ T cells into the TME, elevate IFN-γ levels, decrease PD-1 expression, and reduce the number of exhausted PD-1^+^ T cells [96]. Furthermore, inhibiting IL-1β can increase the percentage of activated cytotoxic T-cells, leading to improved tumor suppression [97].

In addition, C-reactive protein (CRP), a clinical marker of inflammation associated with increased lung cancer risk [98] and progression [98], has been shown to have an interesting relationship with ICIs [98,99]. Given the association between chronic inflammatory states and lung cancer, and the role of IL-1β as an upstream activator of a wide range of inflammatory cytokines and tumorigenesis, CRP may serve as a potential predictive biomarker of response [26]. A study of advanced NSCLC found that elevated CRP levels before treatment with ICIs were predictive of worse PFS and OS along with a lower overall response rate (ORR) [99]. Moreover, high levels of pretreatment CRP were associated with worse PFS and OS, and more rapid increases in CRP levels during treatment were strong predictors of higher risk progressive disease [99]. In the multivariable analysis, there was an association between elevated baseline CRP and lower odds of response (adjusted odds ratio (OR) per doubling of baseline CRP = 0.66, 95% CI: 0.47–0.91, *p* = 0.011). When comparing the baseline and 8-week CRP levels, an early decline of CRP after the initiation of ICIs, defined as a change of 15.6% in the level of CRP, was associated with improved PFS [99]. Thus, validated serum biomarkers are needed to predict response to combination therapy with IL-1β and PD-1/PD-L1 inhibitors.

Despite these potential biomarkers of response to ICIs, there remains room for the further identification of robust markers of response to ICIs. Recently, Assaf et al. demonstrated the clinical significance of circulating tumor DNA (ctDNA) in predicting OS and even response to therapy including ICIs [100]. This study utilized the NSCLC patients from the IMpower150 and the OAK trials and were able to risk stratify patients based on the level of ctDNA [100]. Interestingly, the role of ctDNA levels may be further applied to IL-1β inhibitors. Wong et al., in a molecular analysis of the patients from the CANTOS cohort who developed lung cancer, observed that detectable ctDNA levels were associated with earlier lung cancer diagnosis compared to patients without detectable ctDNA levels [101]. Thus, further studies on the role of ctDNA levels in guiding NSCLC, specifically in the setting of individual or combinatorial inhibition of the IL-1β/PD-1/PD-L1 pathway, are warranted.

## 7. Future Perspective and Direction Targeting IL-1β and PD-1/PD-L1 Pathways in NSCLC

Combining inhibitors targeting IL-1β and PD pathway components may have synergistic effects, enhancing anti-tumor immune responses and overcoming treatment resistance. Future research should focus on identifying optimal combinations, dosage regimens, and treatment schedules to maximize therapeutic efficacy. Developing reliable biomarkers to identify patients who are most likely to benefit from IL-1β and PD pathway targeting is crucial. This would enable personalized treatment strategies, sparing patients from ineffective therapies and reducing unnecessary side effects.

Investigating the underlying mechanisms of resistance and exploring novel strategies to overcome it will be instrumental in improving patient outcomes. Combination therapies, immune checkpoint modulation, and targeted interventions could play a pivotal role in tackling resistance mechanisms. Integrating IL-1β and PD pathway targeting with other immunotherapeutic approaches, such as immune checkpoint inhibitors or adoptive cell therapies, may amplify anti-tumor immune responses and improve overall treatment responses. Preclinical and clinical studies should evaluate the safety and efficacy of these combination approaches.

Translational Research: Bridging the gap between preclinical research and clinical applications is crucial. Conducting well-designed clinical trials to evaluate the efficacy, safety, and long-term outcomes of IL-1β and PD pathway targeting strategies will provide valuable insights and guide future treatment approaches.

MicroRNAs (miRNAs) play a crucial role in oncogenesis, regulating cellular processes like proliferation, differentiation, cell cycle, and apoptosis [102]. High levels of the Bcl-2 protein are linked to advanced tumor stage, lymph node involvement, and distant metastasis. Mutations in the KIT gene are associated with increased galectin levels, while NSCLC patients with brain tumor tissue are more likely to have KIT gene mutation [103]. Future investigations should focus on understanding the molecular mechanisms of miRNA regulation, functional implications of Bcl-2 dysregulation, specific galectins affected by KIT gene mutations, and reasons for the KIT–NSCLC–brain tumor association, offering potential insights for cancer biology and therapeutic targets.

## 8. Conclusions

Both preclinical and clinical studies suggest that the IL-1β/PD-1/PD-L1 pathway plays a crucial role in carcinogenesis of NSCLC and interacts within the TME. The IL-1β/PD-1/PD-L1 pathway may be associated with resistance to therapy and represents a novel therapeutic target for NSCLC. However, further studies are warranted to fully elucidate the underlying mechanisms and determine the optimal combination regimens. As research into the IL-1β pathway continues to progress, it is hoped that the development of new therapeutic strategies will lead to standardized prediction models and biomarkers of response, and ultimately, improved outcomes for patients with NSCLC.

## Figures and Tables

**Figure 1 ijms-24-11547-f001:**
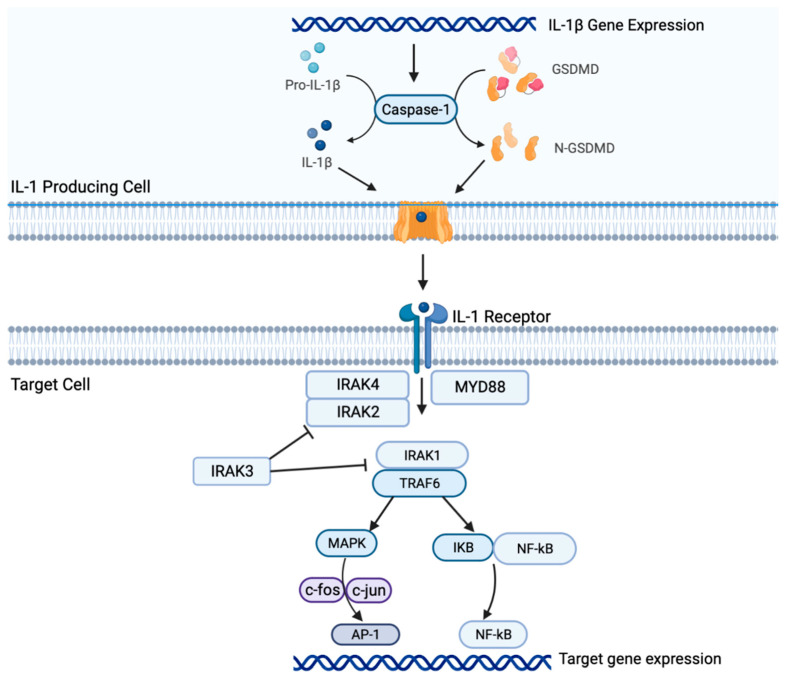
Mechanism of the IL-1β pathway and its downstream effects. Legend: GSDMD—gasdermin D; IRAK—IL-1 receptor-associated kinase; MYD88—myeloid differentiation primary response 88; TRAF6—tumor necrosis factor receptor associated factor 6; MAPK—mitogen-activated protein kinase; IkB—inhibitor of nuclear factor kappa B; NF-kB (nuclear factor kappa B); AP-1—activator protein 1.

**Figure 2 ijms-24-11547-f002:**
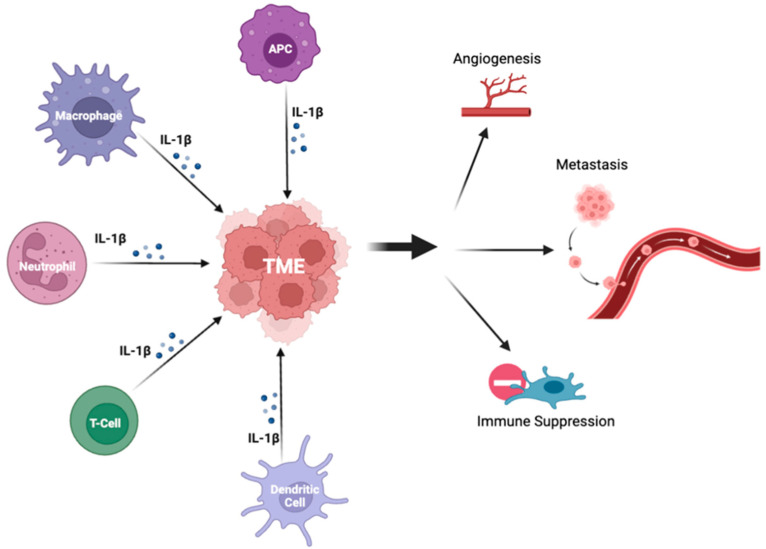
The effects of IL-1β on the tumor microenvironment. The figure depicts immunoregulatory cells altering the TME via secretion of IL-1β, which promote protumor changes including angiogenesis, immune suppression, and metastasis. Legend. TME—tumor microenvironment; APC—antigen-presenting cell; IL-1β—interferon 1-beta.

**Figure 3 ijms-24-11547-f003:**
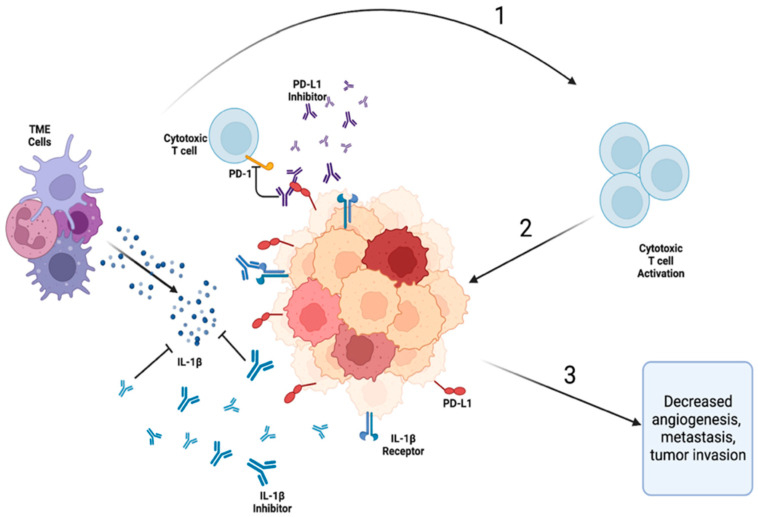
The proposed synergistic effect of co-inhibition of the IL-1β and PD-L1/PD-1 pathways. (1) Inhibition of IL-1β and PD-L1/PD-1 via monoclonal antibodies (inhibitors); (2) activation of cytotoxic T cells because of inhibition of IL-1β and PD-L1/PD-1 pathways; (3) anti-tumor immune response leading to decreased tumor metastasis and growth. Legend. TME—tumor microenvironment; PD-1—programmed cell death 1; PD-L1—programmed cell death ligand 1.

**Table 1 ijms-24-11547-t001:** The summary of phase II/III clinical trials of IL-1β inhibitors in patients with NSCLC.

IL-1β Inhibitors	Type	Target	Trials	Phase, Progress	Experimental Arms	Primary Outcome	Results of the Primary Outcome
Canakinumab	human IgGκ mAb	IL-1β	NCT03968419 (CANOPY-N)	Phase II, completed	canakinumab +/− pembrolizumab; pembrolizumab monotherapy	MPR	MPR: 2.9% (CAN)/17.1% (CAN + PEM)/11.1% (PEM), does not meet the primary endpoint
			NCT03626545 (CANOPY-2)	Phase III, completed	docetaxel +/− canakinumab	OS and incidence of DLTs	Median OS: 10.5 m (CAN) /11.3 m (Placebo) (HR 1.06 95% CI, 0.76–1.48); does not meet the primary endpoint
			NCT03631199 (CANOPY-1)	Phase III, active, not recruiting	pembrolizumab + platinum-based doublet chemotherapy +/− canakinumab	OS, PFS, and incidence DLTs	Median OS: 20.8 m (CAN) /20.2 m (Placebo) (HR 0.87, 95% CI, 0.70–1.10; one-sided *p* = 0.123); median PFS: 6.8 m for both treatment arms (hazard ratio (HR), 0.85; 95% CI, 0.67–1.09; *p* = 0.1); does not meet the primary endpoint
			NCT03447769 (CANOPY-A)	Phase III, terminated	Canakinumab (Canakinumab versus Placebo	DFS	Median DFS: 35 m (CAN)/ 29.7 m (Placebo) (HR 0.94; 95% CI 0.78–1.14; one-sided *p* = 0.258); does not meet the primary endpoint
			NCT04905316 (CHORUS)	Phase II, recruiting	Canakinumab + Chemoradiation + Durvalumab (Single-arm, prospective, phase I/II study)	PFS	Recruiting, no result yet

Legend: mAb—monoclonal antibody; MPR—major pathologic response; DLT—dose-limiting toxicity; DFS—disease-free survival; PFS—progression-free survival; OS—overall survival; m—months.

**Table 2 ijms-24-11547-t002:** The summary of clinical trials of IL-1 receptor antagonists and anti-IL-1R-antibodies related to lung cancer.

IL-1 Inhibitors	Type	Target	Trials	Phase, Progress	Experimental Arms	Primary Outcome	Results of the Primary Outcome
Anakinra	IL-1 receptor mAb	IL-1 receptor	NCT01624766	Phase I, completed	Everolimus (mTOR Inhibitor) + Anakinra/Denosumab	Incidence of AEs, MTD of everolimus	No result published yet
**IL-1R inhibitors**	**Type**	**Target**	**Trials**	**Phase, Progress**	**Experimental Arms**	**Primary Outcome**	**Results of the Primary Outcome**
Nadunolimab (CAN04)	First-in-class fully humanized and ADCC enhanced mAb	IL1RAP	NCT03267316 (CANFOUR)	Phase I/II, Recruiting	CAN04 +/− standard of care treatment	Incidence of Treatment-Emergent AE (Safety and Tolerability)	Infusion-related reactions (41%), fatigue (32%), constipation (27%), diarrhea (27%), decreased appetite (23%), nausea (23%), and vomiting (23%); ORR 53%
			NCT05116891 (CESTAFOUR)	Phase I/II, active, not recruiting	CAN04 + chemotherapy (mFOLFOX or DTX or G/C)	ORR, frequency, duration, and severity of AEs	No result yet
			NCT04452214 (CIRIFOUR)	Phase I, active, not recruiting	CAN04 + pembrolizumab +/− carboplatin and pemetrexed	TEAEs, DLTs, SAEs	No result yet

Legend: MTD—maximum tolerated dose; AE—adverse event; TEAE—treatment-emergent adverse event; SAE—serious adverse event; ORR—overall response rate.

## Data Availability

This original research has not been submitted elsewhere, is not under review by another journal, and has not been published previously.

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
