# Peer review of "Comprehensive Review: Unveiling the Pro-Oncogenic Roles of IL-1ß and PD-1/PD-L1 in NSCLC Development and Targeting Their Pathways for Clinical Management"

_ijms, 2023, doi:10.3390/ijms241411547_

Round 1

Reviewer 1 Report

Castillo and colleagues present a comprehensive review, addressing the pro-oncogenic roles of IL-1ß and PD-1 /PD-L1 in the development of NSCLC and potentials of targeting their associated pathways in the clinical management of non-small cell lung cancer (NSCLC).  The manuscript I is clearly articulated and will serve as an additional reference to existing reviews in this field.

Specific comments

Title. The current title does not reflect the major impact of the data covered in this manuscript. The major coverage surrounds the etiological relevance and potentials of therapeutic targeting the IL-1ß and PD-1 /PD-L1 pathways in NSCLC. Of note, the mechanistic association between IL-1ß and PD-1 /PD-L1 pathways in NSCLC has been widely investigated and hence, not (scientifically) intriguing anymore. Kindly revise the title as appropriate.  

Abstract. Line 32-33 – here, the authors proposed to discuss the role and therapeutic potential of IL-1ß without including PD-1 /PD-L1, which are also major elements of this manuscript as mentioned in line 65-69. Essentially, the objective (s) of this review should be identical under Abstract and Introduction.  

IL-1ß signaling in NSCLC. It is mandatory to categorize this section accordingly.  Aside from the different biological roles of IL-1ß in the progression of NSCLC and resistance to treatment as well as promotion of angiogenesis and chronic inflammation, there is also substantial discussion of PD-1/PD-L1`s role in TME including resistance to TKIs. For better overview, a separate section for PD-1/PD-L1 is essential.

Complex networking in the IL-1ß and PD-1 /PD-L1 pathway. There is indeed an association/networking between IL-1ß and PD-1 /PD-L1, ensuing pro-oncogenic effects in NSCLC. More importantly, this section covers the impact of targeting these pathways that may improve the current treatment modalities in NSCLC. In this context, authors may wish to revise the title of this section accordingly. Equally important, a separate section for the clinical trials on page 9 is recommended.

Line 249-250. 17% of NSCLCs demonstrate type TME. What type of TME based on the 4 categories mentioned in line 247-248 is being quoted?   

Biomarkers of response. This is insufficient - what response is being referred to? Authors may wish to consider line 350-351.

Author Response

Castillo and colleagues present a comprehensive review, addressing the pro-oncogenic roles of IL-1ß and PD-1 /PD-L1 in the development of NSCLC and potentials of targeting their associated pathways in the clinical management of non-small cell lung cancer (NSCLC).  manuscript I is clearly articulated and will serve as an additional reference to existing reviews in this field.

Title. The current title does not reflect the major impact of the data covered in this manuscript. The major coverage surrounds the etiological relevance and potentials of therapeutic targeting the IL-1ß and PD-1 /PD-L1 pathways in NSCLC. Of note, the mechanistic association between IL-1ß and PD-1 /PD-L1 pathways in NSCLC has been widely investigated and hence, is not (scientifically) intriguing anymore. Kindly revise the title as appropriate.  

We appreciate your input. Here is a revised title: “Comprehensive Review: Unveiling the Pro-Oncogenic Roles of IL-1ß and PD-1/PD-L1 in NSCLC Development and Targeting Their Pathways for Clinical Management.”

Abstract. Line 32-33 – here, the authors proposed to discuss the role and therapeutic potential of IL-1ß without including PD-1 /PD-L1, which are also major elements of this manuscript as mentioned in line 65-69. Essentially, the objective (s) of this review should be identical under Abstract and Introduction.  

Line 65-69 has been modified to the following: Recent advances in the understanding of the interplay between inflammation and oncogenesis, specifically involving the IL-1β/PD-1/PD-L1 pathway have been applied to patient care and clinical outcomes. Targeting such pathways in cancers such as NSCLC have the potential to shift cancer management, specifically in patients without targetable genetic mutations; however, there are aspects of this pathway in need of further understanding, especially in the realm of treatment resistance. Therefore, in this review, we aim to discover the role of IL-1β in NSCLC, its involvement in inflammatory pathways and crosstalk with the PD-1/PD-L1 pathway, and explore its potential role in the treatment of NSCLC. 

IL-1ß signaling in NSCLC. It is mandatory to categorize this section accordingly.  Aside from the different biological roles of IL-1ß in the progression of NSCLC and resistance to treatment as well as promotion of angiogenesis and chronic inflammation, there is also substantial discussion of PD-1/PD-L1`s role in TME including resistance to TKIs. For better overview, a separate section for PD-1/PD-L1 is essential.

We value this input; a separate section has been created focusing on the discussion of PD-1/PD-L1’s role in TME including resistance to TKIs. This section is titled “Interplay of PD-1/PD-L1 with TME and their Role in Treatment Resistance.”

Complex networking in the IL-1ß and PD-1 /PD-L1 pathway. There is indeed an association/networking between IL-1ß and PD-1 /PD-L1, ensuing pro-oncogenic effects in NSCLC. More importantly, this section covers the impact of targeting these pathways that may improve the current treatment modalities in NSCLC. In this context, authors may wish to revise the title of this section accordingly. Equally important, a separate section for the clinical trials on page 9 is recommended.

We have revised the title of this section to: “Complex networking of IL-1ß and PD-1/PD-L1 pathway and implications for targeted therapies.”

Another section has been separated with the subtitle:  IL-1β/PD-1/PD-L1 pathways in recent and ongoing NSCLC clinical studies.

Line 249-250. 17% of NSCLCs demonstrate type TME. What type of TME based on the 4 categories mentioned in line 247-248 is being quoted?   

Type 2 TME; we have included this detail in the revised manuscript.

Biomarkers of response. This is insufficient - what response is being referred to? Authors may wish to consider line 350-351.

We have revised the title of this section to “The unmet need for predictive biomarkers of Response and resistance in NSCLC” to clarify this section and its detail.

Reviewer 2 Report

The manuscript entitled “Breaking Barriers in Lung Cancer Research: The Intriguing 1 Connection between IL-1β and PD-1/PD-L1 Pathways” is well written and have complete discussion for the role of IL-1β in in treatment of NSCLC, and its involvement in inflammatory pathways. I have few recommendations to improve the quality of the manuscript.

1.     Change the title “Breaking Barriers in Lung Cancer Research” which does not looks appropriate and does not reflect the theme as mentioned in the abstract.

2.     Abstract does not have details about the sections described in the manuscript. Authors just provide one line “we discuss the role of IL-1β in NSCLC, its involvement in inflammatory path- 32 ways, and explore its potential role in the treatment of NSCLC.”. Hence, they need to explain what authors have discussed in the manuscript and what is the novelty of this review.

3.     Include the citations and discuss more in detail about the role of other factors like miRNAs, BCl-2, and c-Kit in anti-tumor activity.

Author Response

  1. Change the title “Breaking Barriers in Lung Cancer Research” which does not looks appropriate and does not reflect the theme as mentioned in the abstract.

We appreciate your input. Here is a revised title: “Comprehensive Review: Unveiling the Pro-Oncogenic Roles of IL-1ß and PD-1/PD-L1 in NSCLC Development and Targeting Their Pathways for Clinical Management.”

  1. Abstract does not have details about the sections described in the manuscript. Authors just provide one line “we discuss the role of IL-1β in NSCLC, its involvement in inflammatory path- 32 ways, and explore its potential role in the treatment of NSCLC.”. Hence, they need to explain what authors have discussed in the manuscript and what is the novelty of this review.

Here is the revised Abstract to foreshadow the details of the manuscript and share its novelty:

Recent advancements in our understanding of the intricate relationship between inflammation and oncogenesis, particularly involving the IL-1β/PD-1/PD-L1 pathway, have shed light on their application in lung cancer development and clinical outcomes of patients. Targeting these pathways in cancers like NSCLC holds immense potential to revolutionize cancer treatment, particularly for patients lacking targetable genetic mutations. However, despite these promising prospects, there remain certain aspects of this pathway that require further investigation, particularly regarding treatment resistance.

Therefore, the objective of this review is to delve into the role of IL-1β in NSCLC, its participation in inflammatory pathways, and its intricate crosstalk with the PD-1/PD-L1 pathway. Additionally, we aim to explore the potential of IL-1β as a therapeutic target for NSCLC treatment.

  1. Include the citations and discuss more in detail the role of other factors like miRNAs, BCl-2, and c-Kit in anti-tumor activity.

We appreciate the reviewer’s thoughtful comments and recognize the importance of miRNAs, BCl-2, and c-Kit in the field of anti-tumor research in lung cancer. Although we have not directly addressed these factors in our study, we recognize their relevance and potential for further investigation. Additional paragraphs have been added.

MicroRNAs (miRNAs) play a crucial role in oncogenesis, regulating cellular processes like proliferation, differentiation, cell cycle, and apoptosis. High levels of the Bcl-2 protein are linked to advanced tumor stage, lymph node involvement, and distant metastasis. Mutations in the KIT gene are associated with increased galectin levels, while NSCLC patients with brain tumor tissue are more likely to have KIT gene mutations. Future investigations should focus on understanding the molecular mechanisms of miRNA regulation, functional implications of Bcl-2 dysregulation, specific galectins affected by KIT gene mutations, and reasons for the KIT-NSCLC-brain tumor association, offering potential insights into cancer biology and therapeutic targets.

Reviewer 3 Report

This review is entitled Breaking barriers in lung cancer research: the intriguing connection between IL-1beta and PD-1/PD-L1 pathways’. The review ís well written and the content is relevant for the journal. However, there are some minor and major issues that need to be addressed prior to the acceptance.

Minor: 1) Please ensure all abbreviations are listed in the list of abbreviations

             2) Please include references for the figures, if applicable.

Major: 1) Authors should consider to include table(s) to summarize recent advances in vivo targeting IL-1beta pathway to treat lung cancer.

            2) Authors should discuss in detail the interaction between IL-1beta with immune cells and their contribution towards lung cancer treatment.

             3) It will be interesting to include the role of IL-1beta and PD pathway in targeting lung cancer resistance development and its potential treatment, if applicable.

             4) The authors should include the future perspective and direction of targeting IL-1beta and PD pathway for lung cancer treatment.

Author Response

This review is entitled Breaking barriers in lung cancer research: the intriguing connection between IL-1beta and PD-1/PD-L1 pathways. The review is well written and the content is relevant for the journal. However, there are some minor and major issues that need to be addressed prior to the acceptance.

Minor:

1) Please ensure all abbreviations are listed in the list of abbreviations

The list of abbreviations has been updated to include all abbreviations in the manuscript.

2) Please include references for the figures, if applicable.

Given that the figures are lists of clinical trials with NCT numbers, we have elected to include the specific NCT numbers for each trial in the figures, specifically due to the ongoing nature of trials.

Major:

1) Authors should consider including table(s) to summarize recent advances in vivo targeting IL-1beta pathway to treat lung cancer.

Please see Lines 303-313 where several authors/ articles by Kaplanov et al., Li et al., and Jayaraman et al. are cited to describe the in vivo findings targeting the IL-1beta pathway in NSCLC. As the focus of this review is on the interconnectedness and combinatorial targeting of the IL-1beta/PD-1/PD-L1 pathway, we have highlighted the in vivo trials studying the dynamic changes which occur when either IL-1beta or PD-1/PD-L1 or both are targeted.

2) Authors should discuss in detail the interaction between IL-1beta with immune cells and their contribution towards lung cancer treatment.

In the section ‘Complex networking of IL-1ß and PD-1/PD-L1 pathway and implications for tar-gated therapies; we discussed the proposed synergistic effect of co-inhibition of the IL-1β and PD-L1/PD-1 pathways. 1) Inhibition of IL-1β and PD-L1/PD-1 via monoclonal antibodies (inhibitors); 2) activation of cytotoxic T cells because of inhibition of IL-1β and PD-L1/PD-1 pathways; 3) anti-tumor immune response leading to decreased tumor metastasis and growth.

The recent study conducted by Yuan et al. revealed that blocking the IL-1β pathways resulted in an upsurge in cytotoxic CD8+ T cell infiltration and a reduction in the protumor immunosuppressive response. Additionally, this treatment proved to be highly effective in inhibiting the activation of the NF-κB and STAT3 pathways [95].

3) It will be interesting to include the role of IL-1beta and PD pathway in targeting lung cancer resistance development and its potential treatment, if applicable.

We value this input; a separate section has been created focusing on the discussion of PD-1/PD-L1’s role in TME including resistance to TKIs. This section is titled “Interplay of PD-1/PD-L1 with TME and their Role in Treatment Resistance.”

The section titled ‘ Complex networking of IL-1ß and PD-1/PD-L1 pathway and implications for targeted therapies’ has been revised to discuss lung cancer resistance and its potential treatment.

4) The authors should include the future perspective and direction of targeting IL-1beta and PD pathways for lung cancer treatment.

Future perspective and direction targeting IL-1β and PD-1/PD-L1 pathways in NSCLC 

 Combining inhibitors targeting IL-1β and PD pathway components may have synergistic effects, enhancing anti-tumor immune responses and overcoming treatment resistance. Future research should focus on identifying optimal combinations, dosage regimens, and treatment schedules to maximize therapeutic efficacy. Developing reliable biomarkers to identify patients who are most likely to benefit from IL-1β and PD pathway targeting is crucial. This would enable personalized treatment strategies, sparing patients from ineffective therapies and reducing unnecessary side effects.

        Investigating the underlying mechanisms of resistance and exploring novel strategies to overcome it will be instrumental in improving patient outcomes. Combination therapies, immune checkpoint modulation, and targeted interventions could play a pivotal role in tackling resistance mechanisms. Integrating IL-1β and PD pathway targeting with other immunotherapeutic approaches, such as immune checkpoint inhibitors or adoptive cell therapies, may amplify anti-tumor immune responses and improve overall treatment responses. Preclinical and clinical studies should evaluate the safety and efficacy of these combination approaches.

Translational Research: Bridging the gap between preclinical research and clinical applications is crucial. Conducting well-designed clinical trials to evaluate the efficacy, safety, and long-term outcomes of IL-1β and PD pathway targeting strategies will provide valuable insights and guide future treatment approaches.
